# Functional Downregulation of PD-L1 and PD-L2 by CpG and non-CpG Oligonucleotides in Melanoma Cells

**DOI:** 10.3390/cancers14194698

**Published:** 2022-09-27

**Authors:** Johannes Kleemann, Katja Steinhorst, Veronika König, Nadja Zöller, Jindrich Cinatl, Deniz Özistanbullu, Roland Kaufmann, Markus Meissner, Stefan Kippenberger

**Affiliations:** 1Department of Dermatology, Venereology and Allergy, University Hospital, Johann Wolfgang Goethe University, D-60590 Frankfurt am Main, Germany; 2Institute for Medical Virology, University Hospital, Johann Wolfgang Goethe University, D-60590 Frankfurt am Main, Germany

**Keywords:** melanoma, checkpoint, PD-L1, PD-L2, oligonucleotide, quadruplex, IFNγ, JAK, STAT, IRF1

## Abstract

**Simple Summary:**

Although metastatic melanoma is still not a curable disease, targeting of immunologically relevant checkpoints represents a turning point in the treatment. Particularly, targeting the interaction between PD-L1 and its referring receptor PD-1 with antibodies has been shown to activate T-cell function abrogating the evasion of tumor cells from immune recognition. Here, we present another approach that interferes with this system by showing that treatment of melanoma cells with oligonucleotides reduces the expression of PD-L1 (and PD-L2) on tumor cells. Specifically, non-CpG-6-PTO, an ODN that forms superstructures known as G-quartets, has been found to inhibit the interferon-γ-induced signaling cascade which fosters PD-L1 expression. These findings suggest a new therapeutic strategy to interfere with one of the most important immune checkpoints.

**Abstract:**

The clinical application of immune checkpoint inhibitors represents a breakthrough progress in the treatment of metastasized melanoma and other tumor entities. In the present study, it was hypothesized that oligonucleotides (ODNs), known as modulators of the immune response, have an impact on the endogenous expression of checkpoint molecules, namely PD-L1 and PD-L2 (PD-L1/2). IFNγ-stimulated melanoma cells (A375, SK-Mel-28) were treated with different synthetically manufactured oligonucleotides which differed in sequence, length and backbone composition. It was found that a variety of different ODN sequences significantly suppressed PD-L1/2 expression. This effect was dependent on length and phosphorothioate (PTO) backbone. In particular, a sequence containing solely guanines (nCpG-6-PTO) was highly effective in downregulating PD-L1/2 at the protein, mRNA and promoter levels. Mechanistically, we gave evidence that ODNs with G-quartet-forming motifs suppress the interferon signaling axis (JAK/STAT/IRF1). Our findings identify a subset of ODNs as interesting pharmacological compounds that could expand the arsenal of targeted therapies to combat the immunological escape of tumor cells.

## 1. Introduction

Metastatic melanoma still is a not curable disease claiming the lives of 55,500 per year [1]. However, recent therapeutic progress gives hope that a transformation into a chronic but manageable disease might be possible. The decades-long era of having only ineffective chemotherapies, with response rates of 5–20% and median survival rates of 6–9 months, has fortunately come to a close. The introduction of small molecules targeting specific mutations has improved the progression-free survival and the overall survival of patients [2]. Particularly, BRAF mutations, which are present in approximately 43% of primary cutaneous melanomas [3], can be targeted by specific inhibitors [4]. Particularly, the combination of BRAF and MEK inhibitors has improved overall survival in BRAF-mutated advanced melanoma and is now approved as a first-line therapy [2]. Mutations in KIT, which are only present in approximately 1–7% of all melanomas and are more frequent in mucosal and acral melanoma, can also be targeted by specific inhibitors with a substantial effect on disease progression, at least in some patients [5]. Moreover, immunotherapeutic strategies targeting immunologic relevant checkpoints mark a turning point in the treatment of metastatic melanoma. Often tumor cells evade from immune recognition by dysregulating T-cell functions, shifting the balance from auto-immunity to tolerance. Therefore, therapeutic strategies aim for activating stimulatory or/and blocking inhibitory receptors on T-cells [6]. Particularly, the latter approach proved clinical success. It is now ten years ago that ipilimumab, a monoclonal antibody that interrupts the binding between cytotoxic T-lymphocyte-associated antigen 4 (CTLA-4) and B7, was shown to unleash T-cell activity, improving the overall survival of patients with metastatic melanoma [7]. Another immunosuppressive mechanism allowing immune evasion is given by the interaction between programmed death-1 (PD-1) expressed on T-cells, activated B cells, activated NK cells and TILs in different tumor types, and PD-L1 and PD-L2 (PD-L1/2), which are expressed on tumor cells and other cells in the tumor microenvironment [6]. Upon PD-1/PD-L1 binding, T-cell proliferation and survival is inhibited, therefore, hampering effective anti-tumor immunity. Consequently, blocking the PD-1/PD-L1 axis by monoclonal antibodies has been tested in large randomized clinical trials with advanced melanoma patients [8,9]. The outcomes demonstrated superiority compared to ipilimumab with lower toxicity, survival rates of approximately 50% after 2 years and response rates of 37–44% [10]. To date, two approved antibodies, pembrolizumab and nivolumab, with similar clinical efficacy are available for the treatment of malignant melanoma [11]. In this paper, we pursue the question of a non-antibody-based approach to interfere with the PD-1/PD-L1 axis, with focus on synthetic oligonucleotides (ODNs).

ODNs are widely used for research and are also explored as promising anti-tumor therapeutics. Based on the early findings of William B. Coley in 1891 who found that injections of a streptococcal extract into inoperable tumors lead to tumor shrinking [12], it took more than 100 years to associate anti-tumor activity with bacterial DNA [13]. Later, it became evident that a prototypical DNA sequence, the CpG motif, is important for stimulating the immune system [14]. As a receptor for non-methylated CpG DNA the Toll-like receptor-9 (TLR-9) was identified, which is located on intracellular membranes, particularly in endosomes [15]. The property of TLR-9 to stimulate specific immune reactions by activating inflammation-like innate responses in immune cells makes TLR-9 a promising target in tumor therapy [16]. A phase 1b study suggested that intratumorally applied TLR-9 agonist, sd-101 (Dynavax), improved melanoma therapy in combination with pembrolizumab [17]. Likewise, preliminary data derived from a phase I/II trial with PD-L1-refractory melanoma (ILLUMINATE 301) showed maturation of myeloid type 1 DCs (mDC1s) in response to an intratumorally administered synthetic TLR-9 agonist (IMO-2125, tilsotolimod) in combination with CTLA-4 blocking. However, upregulation of PD-L1 by malignant cells has also been detected [18,19].

These findings motivate a closer look into the action of CpG DNA on melanoma cells. In the present study, different ODNs with and without CpG motifs were tested on their impact on PD-L1/2. Moreover, ODN backbone composition and molecule length were considered. Surprisingly, CpG ODNs, particularly non-CpG ODNs composed of guanines, offered a strong suppression of PD-L1/2, indicating that ODNs have the capacity to modify checkpoint protein expression on tumor cells.

## 2. Materials and Methods

### 2.1. Reagents

ODNs with phosphorothioate backbones were synthesized and purified by BioSpring GmbH (Frankfurt/Main, Germany), reconstituted in water and stored at −20 °C. Table 1 shows the sequences used. ODNs were given to the cells at the indicated concentration without DNA complexing reagents. IFNγ was purchased from PeproTech (Hamburg, Germany).

### 2.2. Cell Culture

Human melanoma cell lines (A375, SK-Mel-28, SK-Mel-30, SK-Mel-13, JPC298, MML1 and G361) were purchased from the American Type Culture Collection (ATCC; Manassas, VA, USA). The cell lines were maintained in Dulbecco’s modified Eagle’s medium (DMEM; Invitrogen, Karlsruhe, Germany) supplemented with 10% fetal calf serum (FCS; Greiner, Munich, Germany), 2 mM glutamine, 100 μg/mL streptomycin and 100 U/mL penicillin (Gibco/BRL, Karlsruhe, Germany) at 37 °C in an atmosphere of 5% CO_2_ in air. The cell lines were grown to 80% confluence before passaging using 0.25% trypsin/EDTA solution (Invitrogen). All experiments were conducted according to the Declaration of Helsinki Principles and in agreement with the Local Ethic Commission of the faculty of Medicine of the Johann Wolfgang Goethe University (Frankfurt am Main, Germany). The Local Ethic Commission waived the need for consent.

### 2.3. Fluorescence-Activated Cell Sorting Analysis

A375 or SK-Mel-28 cells were preincubated with IFNγ for 1 h, if not otherwise indicated, and then treated with either 4 µM CpG-1-PTO or nCpG-6-PTO for 24 h in the presence of IFNγ. Consecutively, cells were trypsinized and incubated with mouse anti-human PD-L1/APC conjugate (BD PharMingen) or mouse anti-human PD-L2/PE conjugate (BD Pharmingen). Matched isotype controls (mouse IgG1/APC and mouse IgG1/PE both from R&D Systems) were used. Cells were analyzed on an Accuri C6 Plus flowcytometer (BD Biosciences, Heidelberg, Germany).

### 2.4. Western Blot Analysis

Cells were treated as described and then lysed in Strawn buffer (20 mM HEPES [pH 7.5], 150 mM NaCl, 0.2% Triton × 100 and 10% glycerol) supplemented with a protease inhibitor cocktail (Roche, Mannheim, Germany), sonicated and boiled for 5 min, and separated on SDS-polyacrylamide gels. Consecutively, proteins were immunoblotted on a PVDF membrane. The membrane was blocked in blocking buffer (TBS [pH 7.6], 0.1% Tween-20 and 5% nonfat dry milk) for at least 3 h at 4 °C, followed by incubation with the primary antibody (PD-L1 and PD-L2 both from R&D Systems, Wiesbaden, Germany) in TBS (pH 7.6), 0.05% Tween-20 and 5% BSA. Bound primary antibodies were detected using rabbit anti-goat IgG-horseradish peroxidase conjugates (Dako, Frankfurt, Germany) and visualized with the LumiGlo detection system (CST, Frankfurt, Germany). In the case of PD-L2 a tertiary antibody (anti-rabbit IgG-HRP, CST) was used to enhance the signal strength. Equal loading was controlled by anti-GAPDH or anti-actin (Santa Cruz, Biotechnology, Heidelberg, Germany). In order to detect signaling molecules, the following antibodies were used: p-STAT-1 (Tyr701), STAT-1, p-STAT-2 (Tyr 690), STAT-2, p-STAT-3 (Tyr705), STAT-3, p-JAK-1 (Tyr1022/1023), JAK-1, p-JAK-2 (Tyr1008), JAK-2 and IRF-1, all from CST.

### 2.5. DNA Synthesis

A375 cells were cultivated in microwell plates at a density of 0.5 × 10^4^ cells/0.33 cm^2^. Cells were preincubated with IFNγ (20 ng/mL) for 1 h, then CpG-1-PTO or non-CpG-6-PTO was added at the indicated concentrations. Cells without ODNs served as controls. After 24 h the incorporation rate of 5′-bromo-2′-deoxyuridine (BrdU), which was added for the last 16 h, was determined using a commercial enzyme-linked immunosorbent assay (ELISA) kit (Roche, Mannheim, Germany). Briefly, cells were fixed and immune complexes were formed using peroxidase-coupled BrdU antibodies. A colorimetric reaction with tetramethylbenzidine (TMB) as a substrate gave rise to a reaction product measured at 450 nm using a scanning multiwell spectrophotometer (ELISA-Reader ASYS Expert 96, Deelux Labortechnik, Gödenstorf, Germany).

### 2.6. Membrane Integrity

Cell lysis was quantified using the cytotoxicity detection kit (Roche), which is based on the release of lactate dehydrogenase (LDH) from damaged cells. Briefly, cells were seeded in microwell plates as described above, preincubated with IFNγ (20 ng/mL) for 1 h and then treated with CpG-1-PTO or non-CpG-6-PTO at the indicated concentrations for 24 h. As positive control (maximal cell damage), cells were treated with 1% Triton X-100 (Merck, Darmstadt, Germany). Subsequently, the cell-free supernatants were incubated with NAD+, which was reduced by LDH to NADH/H+. In a second step, NADH/H+ reduced a yellow tetrazolium salt to a red-colored formazan salt. The amount of red color was proportional to the number of lysed cells. For quantitation, the absorbance of the reaction product was measured at 490 nm using a multiwell spectrophotometer (see above).

### 2.7. Histone-Associated DNA Fragments

Advanced apoptosis was quantified on the basis of cytoplasmic histone-associated DNA fragments using the cell death detection ELISA kit (Roche) according to the manufacturer’s manual. In brief, cells were cultured in microwell plates as described above, preincubated with IFNγ (20 ng/mL) for 1 h and then treated with CpG-1-PTO or non-CpG-6-PTO at the indicated concentrations for 24 h. As positive control served 1 µM staurosporin. Thereafter, the cytosolic fraction (200 g supernatant) was used as an antigen source in a sandwich enzyme-linked immunosorbent assay with a primary anti-histone antibody coated to a microtiter plate and a secondary anti-DNA antibody coupled to peroxidase. Optical density was measured at 530 nm using a multiwell spectrophotometer (see above).

### 2.8. Real-Time RT-PCR Analysis

To IFNγ preincubated cells (20 ng/mL, 1 h), 4 µM CpG-1-PTO or nCpG-6-PTO was added. After 3, 6 and 24 h, total cellular RNA was isolated using the ExtractMe total RNA kit (Blirt, Gdańsk, Poland). After DNase digestion, a total amount of 25 ng of RNA was used for first-strand cDNA synthesis using the QuantiTect SYBR Green RT-PCR kit (Qiagen, Hilden, Germany). Real-time PCRs were performed on the Light Cycler system 2.0 (Roche). The following primers were used: PD-L1 left: 5′-GGC ATC CAA GAT ACA AAC TCA A-3′, PD-L1 right: 5′-CAG AAG TTC CAA TGC TGG ATT A-3′, PD-L2 left: 5′-ACC TGC CAG GCT ACA GGT TA-3′, PD-L2 right: 5′-AGG AAC GCT GAC GTT TGG-3′, β-actin left: 5′-CAA CCG CGA GAA GAT GAC-3′ and β-actin right: 5′-GTC CAT CAC GAT GCC AGT-3′. The relative expression of transcripts was determined using the 2-∆∆CT method [20].

### 2.9. PD-L1/2 Promoter Transient Reporter Assay

A375 cells were seeded in 48-well plates (1.5 × 10^4^ cells/well) for 18 h and then transfected with 0.3 µg/well of each experimental firefly luciferase construct and 5 ng/well pRL-SV40-Renilla (Promega, Madison, WI, USA) for normalization using the SuperFect transfection reagent (Qiagen, Hilden, Germany) according to the manufacturer’s instruction. Human PD-L1 and PD-L2 reporter gene constructs including deletion mutants for transcription binding sites (STAT-1/STAT-3, STAT-2/STAT-5 and IRF1) were provided by Antoni Ribas [21]. Afterwards, transfection cells were stimulated with 20 ng/mL IFNγ for 1 h and then treated with 4 µM CpG-1-PTO or 4 µM nCpG-6-PTO for 16 h. Untreated cells served as the control. Luciferase activity was measured with the dual-luciferase reporter assay system (Promega) using a luminometer (Lumat LB 9507, Fa. Berthold, Pforzheim, Germany).

### 2.10. Chromatin Immunoprecipitation (ChIP) Assays

ChIP was performed using the Chromatrap ChIP-seq assay (Chromatrap, Norfolk, England). Briefly, confluent cultures (20 cm^2^) were treated with 4 µM ODNs for 1 h and then stimulated with 20 ng/mL IFNγ. After 6 h, proteins were fixed (1% formaldehyde at room temperature for 10 min) and then scraped in ice-cold PBS. After membrane lysis, nuclear extracts were prepared and sonicated (Fa. Bandelin, Berlin, Germany) in order to yield DNA fragments ranging from 100 to 500 bp as controlled by agarose gel electrophoresis. After centrifugation, supernatants were subjected to IP using an IRF1 antibody (CST; for control IgG and antibodies against H3K4me3) and consecutively given to protein-A columns. After elution, the protein/DNA crosslink was reversed under high salt conditions (65 °C, overnight); proteins were digested by proteinase K. The purified and cleaned DNA was used for real-time PCR using an IRF-specific primer set [21]: PD-L1-IRF1 FWD: 5′-GCT TTA ATC TTC GAA ACT CTT CCC-3′; PD-L1 IRF1 REV: 5′-AAG CTG TGT ATA GAA ATG AAA CAG-3′; PD-L2 IRF1α FWD: 5′-CCA AGA GCC AAA TCA GGA ATG-3′ and PD-L2 IRF1α REV: 5′-GCC TGG CTT ATT TGG AAA GTT-3′. The data were presented as mean percent inputs (the relative amount of IP DNA compared to DNA after PCR analysis).

### 2.11. Cellular Uptake, Quadruplex (G4) Formation and Binding to the IFNGR1

To examine the uptake of ODNs, A375 cells were placed on glass coverslips in the presence of 4 µM 5′-Cy3-labelled CpG-1-PTO and nCpG-6-PTO for 24 h. Consecutively, cells were examined using a BZ-X800 fluorescence microscope (Keyence, Neu-Isenburg, Germany). Nuclei were stained with DAPI. Formation of G4 was tested as described [22,23]. Briefly, 0.2 µg 5′-Cy5-labelled ODNs were mixed with 200 and 400 ng of the G4-specific antibody BG4 (Biozol ABA-AB00174-1.1, Eching, Germany) for 15 min at room temperature. After separation by 10% non-denaturing PAGE (100 V, corresponding to 14.7 V/cm) with 0.5× TBE, fluorescence was captured using the LI-COR Odyssey gel documentation system (Bad Homburg, Germany). Likewise, specific interaction with the IFNGR was tested by incubating the Cy3-labelled ODNs with either 200 ng or 400 ng IFNGR1 (R&D Systems) or IFNGR2 (Novus Biologicals, Centennial, CO, USA) before separation by PAGE.

### 2.12. Statistical Analysis

All data were presented as mean values ± standard deviations. Statistical significance of the data was calculated by the ANOVA test (BIAS, Frankfurt, Germany). Each set of data was related to the referring controls. Differences were considered significant at *p* < 0.05, indicated by an asterisk.

## 3. Results

### 3.1. Oligonucleotides Suppress the Protein Expression of PD-L1 and PD-L2 in Melanoma Cells

A set of different ODN sequences, given in Table 1, were tested on the regulation of PD-L1/2 in A375 melanoma cells (Figure 1A, Appendix A: Original western blots).

In order to detect a possible suppression by ODNs, PD-L1/2 expression was induced by a pretreatment with 20 ng/mL IFNγ for 1 h before the addition of ODNs. After 24 h, cells were harvested and subjected to Western blotting. Treatment with IFNγ leads to the described strong upregulation of PD-L1/2, particularly of PD-L1 [21]. As an interesting result, we found that almost all ODNs used offered a suppression of the IFNγ-induced PD-L1/2 expression, although differently pronounced. No suppression was detected for CpG-12-PTO, a deletion mutant of CpG-1-PTO with a centered CpG motif and a total length of 12 nucleotides. In contrast, the original CpG-1-PTO (also known as CpG-B-1826), a B/K-type CpG sequence of 20 nucleotides in length, showed significant suppression of PD-L1 and very slight suppression of PD-L2. Testing of the same sequence with a phosphodiester backbone (CpG-1-PDE) showed almost no attenuation indicating the relevance of the backbone in this context. The inhibitory effect was again restored using the reverse sequence of CpG-1-PTO, named CpG-1-PTO-rev. Moreover, oblimersen (G3139), a sequence originally developed as Bcl-2 antisense [24], showed some suppression. Interestingly, oblimersen also contains a CpG motif which seems to contribute to the anti-cancer response [25]. The ODR nCpG-1-PTO, which has the original CpG-1-PTO sequence with the two CpG motifs substituted by other nucleotides, also showed some suppression but not as strong as the original CpG-1-PTO. Likewise, a random sequence composed of the same nucleotides as CpG-1-PTO called ‘scrambled’ offered some suppression at least on PD-L1. Interestingly, 20mers composed of thymines (nCpG-3-PTO) and cytosines (nCpG-5-PTO) showed a marked suppression. A complete suppression of PD-L1/2 was achieved by nCpG-6-PTO, a 20mer solely composed of guanines.

### 3.2. PD-L1/2 Protein Suppression by CpG-1-PTO and nCpG-6-PTO

From the screening experiment shown in Figure 1A, nCpG-6-PTO was identified as a potent suppressor of PD-L1/2 expression. In the following, we compared the effects of nCpG-6-PTO with CpG-1-PTO, a sequence with two CpG motifs. In order to generate quantitative results, we established a FACS-based detection of surface PD-L1/2. Figure 1B shows exemplary results. The summary of nine experiments using A375 cells is given in Figure 1C. It was found that treatment with 20 ng/mL IFNγ strongly induced PD-L1 (98%) and PD-L2 (60%). In the presence of 4 µM CpG-1-PTO, PD-L1 was reduced to 69% and PD-L2 to 53%; both reductions were statistically significant. Addition of 4 µM nCpG-6-PTO reduced expression levels massively, to 4% for PD-L1 and 5% for PD-L2. Moreover, SK-Mel-28, another human melanoma cell line, was tested (Figure 1D). IFNγ reduced PD-L1 to 97% and PD-L2 to 35%. In the presence of CpG-1-PTO, PD-L1 was reduced to 87%; PD-L2 levels were not significantly changed. With nCpG-6-PTO, PD-L1 was reduced to 17% and PD-L2 to 13%; both reductions were significant. Moreover, different melanoma cell lines were screened for their basal (non-IFNγ-induced) levels of PD-L1/2 (see Appendix A). It was found that a group of cell lines (JPC298, SK-Mel 13, MML-1 and G361) showed moderate basal expression of PD-L1, ranging from 20% (MML-1) to 36% (JPC298) compared to the isotype control. This finding was contrasted by a group of cell lines (A375, SK-Mel 28 and SK-Mel-30) showing strong basal PD-L1 expression higher than 50%, with SK-Mel-28 cells showing the highest expression (89%). Basal PD-L2 expression was significantly lower in all tested cell lines, ranging from 4% (SK-Mel 30) to 19% (MML-1). In the following, it was tested if constitutive PD-L1 expression was susceptible to ODN treatment (see Appendix A). This was tested in the cell lines with high basal expressions (A375, SK-Mel-28 and SK-Mel-30). CpG-1-PTO showed no reduction in the basal levels; nCpG-6-PTO reduced PD-L1 levels in all cell lines; in A375 and SK-Mel-28 the results were statistically significant. With regard to PD-L2, cell lines with relatively high levels of PD-L2 (A375, SK-Mel-28 and MML-1) were examined (see Appendix A). Although there was found some suppression of PD-L2, the data failed to be statistically significant. In summary, these results showed a partial suppression of PD-L1 by nCpG-6-PTO, also under conditions without IFNγ.

### 3.3. PD-L1/2 Protein Suppression by CpG-1-PTO and nCpG-6-PTO: Dependence on Concentration and Molecule Length

In order to learn more about the effect of CpG-1-PTO and nCpG-6-PTO, concentration dependency was tested in A375 (Figure 2A,B) and SK-Mel-28 (Figure 2C,D) cell lines.

From data shown in Figure 1, we learnt that nCpG-6-PTO had a far higher efficiency to suppress PD-L1/2 than CpG-1-PTO. Therefore, different concentration ranges were chosen for both ODNs, namely 4, 6 and 8 µM for CpG-1-PTO and 1, 2 and 4 µM for nCpG-6-PTO. For CpG-1-PTO, we found a concentration dependent inhibition of PD-L1 in A375 (Figure 2A) and SK-Mel-28 (Figure 2C) cells. However, the effect was more pronounced in A375 cells. The effects on PD-L2 expression were diverse: a small but statistically significant suppression with increasing concentration was detected in A375 cells (Figure 2A) and only a small suppression in SK-Mel-28 cells at 8 µM was detected (Figure 2C). The ODN nCpG-6-PTO, although given in lower concentrations, showed very distinct suppression of PD-L1/2 in A375 (Figure 2B) and SK-Mel-28 (Figure 2D) cells, with high statistical significance. Common to CpG-1-PTO is that relative PD-L1 suppression was more pronounced than relative PD-L2 suppression, and the response in A375 cells was stronger than in SK-Mel-28 cells. In the next section, we investigated if the above-described effect was dependent on molecule length. Therefore, deletion mutants of CpG-1-PTO and nCpG-6-PTO (see Table 1) were tested in A375 (Figure 2E,F) and SK-Mel-28 (Figure 2G,H) cells. In general, it was found that with decreasing molecule lengths, the suppressive effect on PD-L1/2 vanished. However, there were quantitative differences. CpG-9-PTO, a 6mer of CpG-1-PTO, lost its ability to suppress PD-L1 in A375 cells (Figure 2E); in SK-Mel-28 cells, the 12mer CpG-12-PTO already offered no suppressive effect (Figure 2G). In regard to PD-L2, sequences shorter than 16 nucleotides (CpG-14-PTO) showed no suppression in A375 cells (Figure 2E). In SK-Mel-28 cells, all deletion mutants including the original CpG-1-PTO offered no suppression (Figure 2G). Although deletion mutants of nCpG-6-PTO lost some of their suppressive properties, the effect was less pronounced compared to CpG-1-PTO mutants. Even the 6mer nCpG-6G-PTO showed significant PD-L1 suppression in A375 (Figure 2F) and SK-Mel-28 (Figure 2H) cells. Similarly for PD-L2, all sequences longer than 6mer ODNs showed distinct suppression in both cell species.

Moreover, we tested the impact of different application protocols by varying the preincubation time of the IFNγ (see Appendix A) and the ODNs (see Appendix A). It was found that PD-L1 expression increased slightly with the extension of the IFNγ preincubation time (1 h, 6 h and 24 h); the impact on PD-L2 was more distinct. Both ODNs showed the above-described PD-L1/2 suppression in all preincubation settings. Moreover, after 24 h preincubation with IFNγ, the ODNs, in particular nCpG-6-PTO, showed significant PD-L1 suppression. In another experiment, the ODN preincubation time was extended (1 h, 3 h and 6 h) before IFNγ was added (see Appendix A). The ODNs showed suppression of PD-L1/2 which was not dependent on the preincubation time.

### 3.4. CpG-1-PTO and nCpG-6-PTO Suppress PD-L1/2 mRNA and Promoter Activation

As both tested melanoma cell lines offered qualitatively similar behavior in regard to IFNγ treatment and responsiveness towards both ODNs, we focused the following on A375 cells, in which the observed effects were more distinctive. Next, regulation of PD-L1/2 gene transcription by CpG-1-PTO and nCpG-6-PTO was investigated by real-time RT-PCR (Figure 3).

A375 cells were prestimulated with IFNγ for 1 h and then treated with 4 µM CpG-1-PTO or 4 µM nCpG-6-PTO for 3, 6 or 24 h. The PD-L1 expression induced by IFNγ rose 2.7-fold after 3 h, 5.5-fold after 6 h and 63.2-fold after 24 h (Figure 3A). Both CpG-1-PTO and nCpG-6-PTO inhibited the IFNγ-induced upregulation at all time points. Particularly, nCpG-6-PTO completely reversed the IFNγ effect. The upregulation of PD-L2 by IFNγ was markedly lower, reaching a level of 2.69-fold after 24 h compared to the untreated control (Figure 3B). Both ODNs reduced the IFNγ-induced PD-L2 upregulation with nCpG-6-PTO being more potent than CpG-1-PTO. Moreover, PD-L1/2 promoter transactivation assays were performed (Figure 4A,B).

IFNγ induced a strong PD-L1 activation of more than 25-fold in the full-length construct and in the ΔSTAT-1/3 deletion mutant (Figure 4A). In the presence of CpG-1-PTO and nCpG-6-PTO, promoter activation was significantly lessened in the full-length construct. Moreover, in the ΔSTAT-1/3 deletion mutant, both ODNs reverted the IFNγ induction, with nCpG-6-PTO being more potent than CpG-1-PTO. However, the latter failed statistical significance (*p* = 0.089). In another deletion mutant devoid of ΔSTAT-2/5 binding sites, activation by IFNγ amounted to 5-fold compared to the negative control. This level was reduced by both ODNs, again nCpG-6-PTO being stronger than CpG-1-PTO; however, the measured reductions were not statistically significant (CpG-1-PTO, *p* = 0.828; nCpG-6-PTO, *p* = 0.695). A deletion of the IRF1 binding site completely abrogated the activation by IFNγ as described [21]. Figure 4B shows the PD-L2 promoter transactivation in the wild-type and deletion mutants. Upon IFNγ exposure, a 6.7-fold activation was observed in the full-length construct. Both ODNs significantly abrogated this effect, nCpG-6-PTO more distinctively than CpG-1-PTO. Deletion of the STAT-3 binding site reduced the basal activation level. In this construct, IFNγ led to a 16.6-fold increase which became diminished by both ODNs, with nCpG-6-PTO showing a statistically significant reduction. Moreover, constructs harboring deletions for two putative IRF binding sites (IRFα and IRFβ) were investigated. Deletion of IRF1α drastically reduced PD-L2 reporter expression upon IFNγ. In contrast, deletion of the IRF1β site amplified the IFNγ response 16.5-fold, pointing to an inhibitory effect of this site. The IFNγ response was more than halved by CpG-1-PTO and even stronger for nCpG-6-PTO. Double mutations of STAT-3 and IRFα completely abrogated the IFNγ response. The above-described promoter studies confirmed the essential role of IRF1 for the IFNγ-induced activation of PD-L1 and PD-L2 promoters. In the following, it was tested whether the observed promoter suppression by CpG-1-PTO and nCpG-6-PTO was due to decreased IRF binding. Hence, cells were pretreated with ODNs and then stimulated with IFNγ before conducting ChIP assays using IRF antibodies for precipitation. Figure 4C shows the IRF1-dependent upregulation of the PD-L1 promoter at almost 100% compared to the control set at 1%. This data confirmed the relevance of IRF1 as a pivotal transcription factor for the IFNγ-dependent PD-L1 upregulation [21]. Both CpG-1-PTO and nCpG-6-PTO counteracted this upregulation with only nCpG-6-PTO showing statistical significance. The IFNγ/IRF1-dependent upregulation for PD-L2 was approximately 10% (Figure 4D), which was in the range described by others [21]. CpG-1-PTO showed no statistically relevant effect which was possibly due to the generally weak effects of this ODN. In contrast, nCpG-6-PTO halved the IFNγ-mediated upregulation, which was statistically significant.

### 3.5. CpG-1-PTO and nCpG-6-PTO Inhibit Signaling Molecules of the IFN Type I and II Pathway

As shown by others, IFNγ activates type II interferon canonical signaling molecules with typical upregulation of the JAK-1/JAK-2/STAT-1/IRF1 axis. However, an overlap with type I signaling with activation of STAT-2 and STAT-3 was also observed [21]. Therefore, activation of these signaling molecules was examined upon IFNγ in the presence and absence of CpG-1-PTO and nCpG-6-PTO (Figure 5, Appendix A: Original western blots).

At first, time points where responsiveness towards the indicated treatment was at the maximum were evaluated. For the detection of IRF, proteins were extracted after 3 h and 6 h; the other molecules, which were upstream from IRF, were extracted after 10 min and 30 min. It was found that treatment with IFNγ induced significant phosphorylation in canonical and non-canonical signaling molecules such as JAK-1, JAK-2, STAT-1, STAT-2 and STAT-3. In concert with previous findings, massive upregulation of IRF expression was detected [21]. The ODNs alone had no effect on phosphorylation or expression of the investigated proteins. However, the IFNγ-induced levels were significantly decreased by both ODNs; again nCpG-6-PTO showed a very strong suppressive effect.

### 3.6. nCpG-6-PTO Enters the Cell, Forms Quadruplexes (G4) and Binds to the IFNGR2

Cellular uptake of 5′-Cy3-labelled CpG-1-PTO and nCpG-6-PTO into A375 cells was monitored with fluorescence microscopy (Figure 6A).

As displayed, only nCpG-6-PTO was taken up by A375 cells; here, accumulations of distinct clusters were detected in the cytoplasm as well as outside the cells. Of note, no fluorescence was detected after treatment with CpG-1-PTO. In order to exclude a fault in the labelling process or bleaching by improper storage, a 1:10 serial dilution of both ODNs was dotted on a plastic dish and the fluorescence was captured using the LI-COR Odyssey gel documentation system (Figure 6B, Appendix A: Original western blots). Fluorescence within the same range was found for both ODNs. This indicated a sequence-specific uptake in A375 cells. As aforementioned, we hypothesized that secondary structures of the ODN may be important for the observed effects. Therefore, the formation of secondary structures was examined by separating 5′-Cy3-labelled CpG-1-PTO and nCpG-6-PTO by non-denaturing PAGE (Figure 6C, Appendix A: Original western blots). The first lane of each gel showed the native pattern. CpG-1-PTO displayed a condensed low molecular pattern of approximately three bands at the bottom of the gel. In contrast, nCpG-6-PTO featured a widespread ladder-like pattern reaching into the high molecular range. In order to test the presence of G4, both ODNs were mixed with 200 and 400 ng BG4, an antibody specific to G4 [22,23]. Binding of BG4 to G4 was documented by a fluorescence shift to the higher molecular range. Particularly, nCpG-6-PTO showed distinct binding to BG4 indicating the presence of G4. In the case of CpG-1-PTO, only a very small amount of the ODN was retarded by BG4. From Western blot experiments, we learnt that nCpG-6-PTO, the G4-forming ODN, suppressed signaling molecules of the IFN type I and II pathway (Figure 5). Therefore, we tested if nCpG-6-PTO interfered on the level of the IFNGR (Figure 6D). Cy3-labelled ODNs were either mixed with 200 ng or 400 ng IFNGR1 or IFNGR2 before separation by PAGE. Here, we observed a concentration dependent binding of nCpG-6-PTO to IFNGR2; a mixture of nCpG-6-PTO and IFNGR1 showed no fluorescence shift. Experiments with CpG-1-PTO showed no fluorescence shift in combination with either IFNGR1 or IFNGR2. These results suggested that nCpG-6-PTO inhibited receptor signaling was mediated by IFNGR2, while IFNGR1 was the primary site of ligand binding [26].

## 4. Discussion

In this paper, we reported the functional checkpoint inhibition in melanoma cells by DNA fragments (ODNs). Both PD-L1 and PD-L2 are cell membrane-bound glycoproteins sharing a 40% amino acid homology to each other [27]. Despite the similarity, their expression profiles seem different: PD-L1 is predominantly expressed on tumor cells and on tumor infiltrating immune cells (in particular CD3^+^ and CD8^+^), but also in normal human tissue such as myeloid dendritic cells (DCs), macrophages, placental trophoblasts, myocardial endothelium and cortical thymic epithelial cells [28]. PD-L2 expression is less prevalent on tumor tissue compared to PD-L1 but is also detected on normal cells such as DCs, macrophages, placental endothelium and medullary thymic epithelial cells [28]. In contrast to PD-L1, there seems to be no clear correlation between PD-L2 expression and cancer prognosis [29]. In general, the physiological role of PD-L2 is not yet completely clear. It has been suggested that PD-L2 outcompetes PD-L1 from their common receptor, thus, limiting T-cell exhaustion [30]. This suits to a finely balanced regulation between T-cell activation and tolerance [31]. It was a surprise to us that ODNs took part in this regulation. When in 1944, Oswald Avery together with two colleagues first described DNA as a carrier of genetic information [32], it could not be foreseen that DNA has properties reaching far beyond the coding function. These range from storing physical energy by torsional stress, facilitating the transit of DNA and RNA polymerases [33], to forming superstructures such as quadruplexes, which for example provide maintenance of telomeres [34,35]. Moreover, ODNs are widely used for research but are also considered as promising therapeutics. Typically, ODNs are designed to selectively inhibit the translation of disease-associated genes. For this purpose two steps are important: (a) the absorption to the cell membrane and (b) the internalization [36]. The latter can be facilitated by the use of cationic liposomes. In our experiments, no such additives were used. Considering the heterogenic set of sequences used in this study, an antisense effect against PD-L1/2 seemed rather unlikely. Instead, we first suspected a TLR-9-dependent mechanism. TLR-9 is mainly known as a receptor for unmethylated CpG DNA. However, DNA lacking a CpG motif can also activate TLR-9 signaling [15,37]. Although TLR-9 is present in melanoma cells of clinical biopsies [38] and also in in vitro cultured melanoma cells including A375 cells [39], experiments using chloroquine, a compound known to interfere with TLR-9 and its ligand [40], do not reverse PD-L1/2 downregulation (see Appendix A). This indicates a TLR-9-independent mechanism in response to CpG-1-PTO and nCpG-6-PTO.

Instead, we presented evidence that ODNs which form stable secondary structures convey the observed effect. The presence of exclusively guanines in nCpG-6-PTO prompted us to investigate the molecular properties in more detail. It is now well documented that DNA not only forms the canonical duplex helix but can also fold into different types of structures [41]. In particular, the formation of successive G-tetrads (G4) of Hoogsteen hydrogen-bonded guanine bases has been shown to have an impact on genome functions such as transcription, replication and epigenetic regulation [42]. Notably, telomeres are rich in G4 suggesting a functional link to the telomerase function with an impact on oncogenesis [43]. Moreover, synthetic G4-forming ODNs are currently being investigated for therapeutic use. Depending on the sequences used, G4-aptamers recognize proteins such as nucleolin, Shp2 and VEGF, which make them interesting for anti-cancer therapies [44]. For example, the G4-forming ODN AS1411, an aptamer to nucleolin [45], has already been tested in clinical trials against AML (NCT01034410) [46] and RCC (NCT00740441) [47]. The ODN nCpG-6-PTO used in our study also formed stable G4 as detected using BG4, an antibody specific for G4. As the target structure for these aptamers, we identified the IFNγ receptor (IFNGR1/2). Functional relevance was confirmed by showing the suppression of canonical type II and type I signaling molecules. An overlap of both signaling pathways in response to IFNγ was recently described [21]. The inhibition of STAT proteins, in turn, reduces the expression of IRF1, the most important transcription factor in the IFNγ-mediated PD-L1/2 induction [21]. Our results showed a complete disappearance of IRF1 3 h after treatment with nCpG-6-PTO. Confirmatory to this, we detected diminished IRF-1 binding to the DNA (ChIP assay), inhibition of PD-L1/2 promoter transactivation and mRNA expression.

## 5. Conclusions

In summary, these findings introduce a new therapeutic concept to interfere with checkpoint inhibitor besides the use of antibodies. There is decades-long experience with applying DNA-based compounds in patients which helps to estimate important parameters such as pharmacology and toxicity [48]. Our in vitro data, at least, showed no impact on proliferation, LDH release and apoptosis (see Appendix A). Therefore, treatment with ODNs, in particular nCpG-6-PTO, alone and/or in combination with other means, may improve the outcome of melanoma treatment.

## 6. Patents

This section is not mandatory but may be added if there are patents resulting from the work reported in this manuscript.

## Figures and Tables

**Figure 1 cancers-14-04698-f001:**
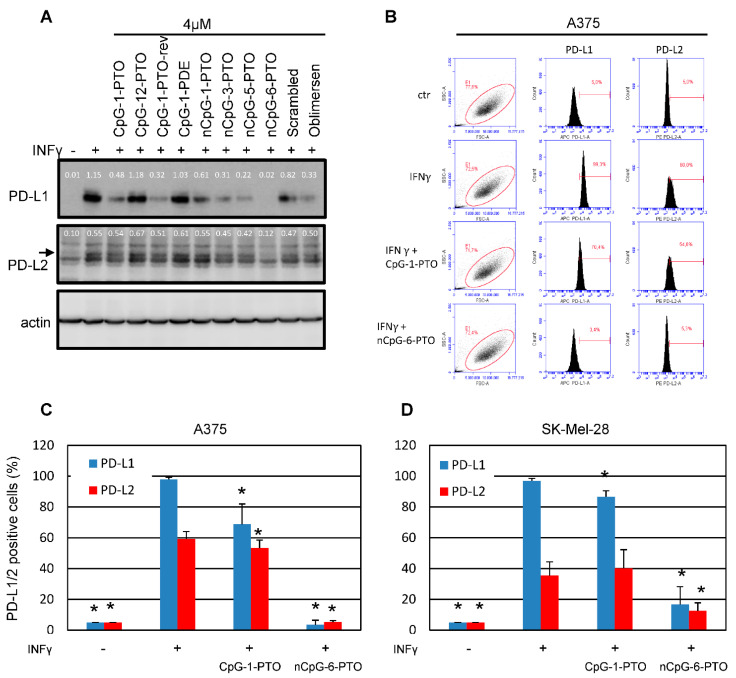
Oligonucleotides suppress the protein expression of PD-L1 and PD-L2. (**A**) A375 melanoma cells prestimulated for 1 h with IFNγ (20 ng/mL) were exposed to different oligonucleotides (4 µM). Oligo characteristics in regard to backbone and sequence are given in Table 1. After 24 h, total protein was extracted and separated by SDS-PAGE. The blotted proteins were probed with anti-PD-L1 and anti-PD-L2. Probing with anti-beta actin served as loading control. The images show representative results. (**B**) Exemplary result of a FACS scan against PD-L1 and PD-L2 after treatment with 4 µM CpG-1-PTO and 4 µM nCpG-6-PTO in A375 cells. Summary of 9 independent FACS experiments using (**C**) A375 and (**D**) SK-Mel-28 cells treated with 4 µM CpG-1-PTO or nCpG-6-PTO. The standard deviations are indicated. Data were related to the referring positive control (IFNγ). * *p* < 0.05.

**Figure 2 cancers-14-04698-f002:**
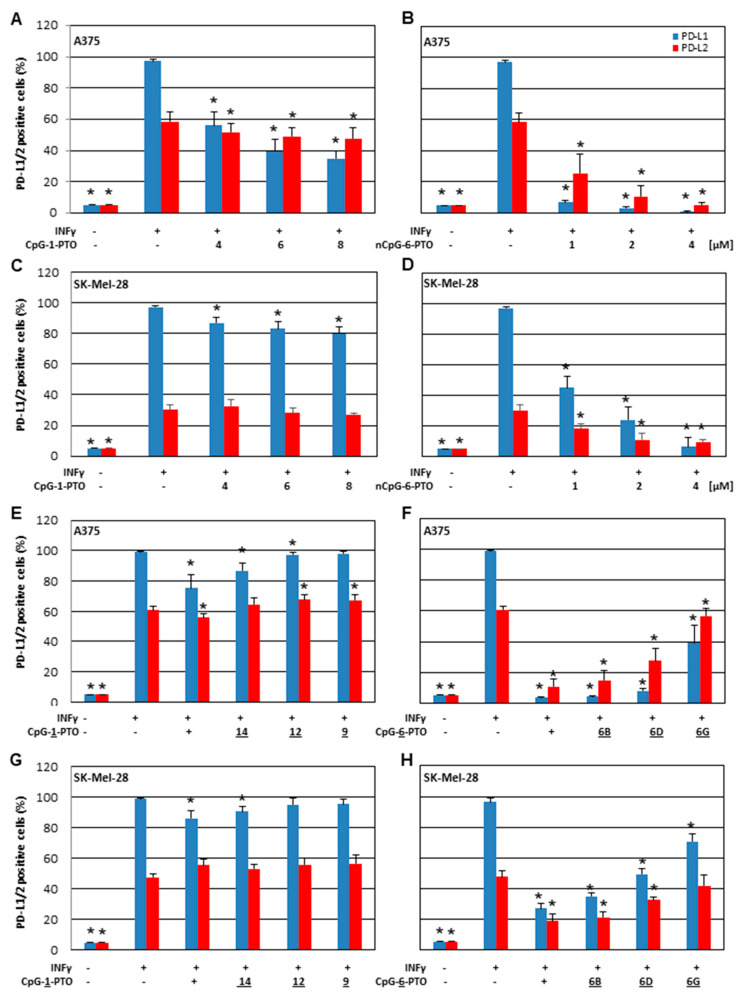
Suppression of PD-L1 and PD-L2 by nCpG-6-PTO and CpG-1-PTO—impact of concentration and molecule length. IFNγ-stimulated A375 cells were treated with increasing concentrations of (**A**) nCpG-6-PTO (1, 2 and 4 µM) or (**B**) CpG-1-PTO (4, 6 and 8 µM). Likewise, SK-Mel-28 cells were treated with increasing concentrations of (**C**) nCpG-6-PTO or (**D**) CpG-1-PTO. Moreover, the impact of ODN length was investigated. IFNγ-stimulated A375 cells were treated with (**E**) 4 µM CpG-1-PTO and the referring deletion mutants (CpG-14/12/9-PTO) or (**F**) 4 µM nCpG-6-PTO and the referring deletion mutants (nCpG-6B/6D/6G-PTO). Likewise, SK-Mel-28 cells were treated with (**G**) CpG-1-PTO or (**H**) nCpG-6-PTO and their referring deletion mutants. After 24 h expression, PD-L1 and PD-L2 were measured by FACS. Each bar represents the mean of 6 independent experiments. The standard deviations are indicated. Data were related to the referring positive control (IFNγ). * *p* < 0.05.

**Figure 3 cancers-14-04698-f003:**
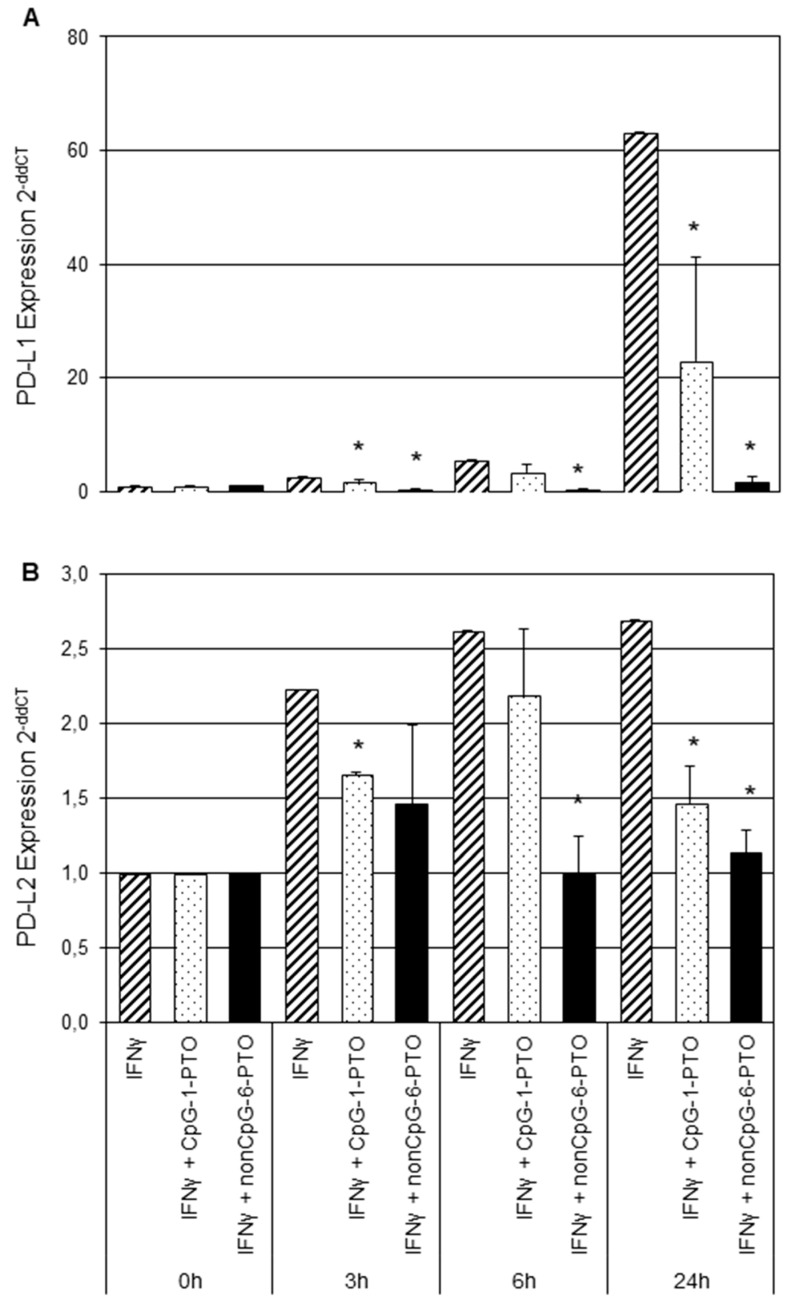
Expression analysis of PD-L1 and PD-L2 by real-time RT-PCR. A375 melanoma cells were stimulated with IFNγ for 1 h, followed by an incubation of either CpG-1-PTO or nCpG-6-PTO for 3, 6 or 24 h. Consecutively, total RNA was extracted, followed by quantitative RT-PCR as described. Results for (**A**) PD-L1 and (**B**) PD-L2 are displayed. Each column shows the mean of three independent experiments. Standard deviations are indicated. Statistical analysis was performed in relation to controls treated with IFNγ solely; * *p* < 0.05.

**Figure 4 cancers-14-04698-f004:**
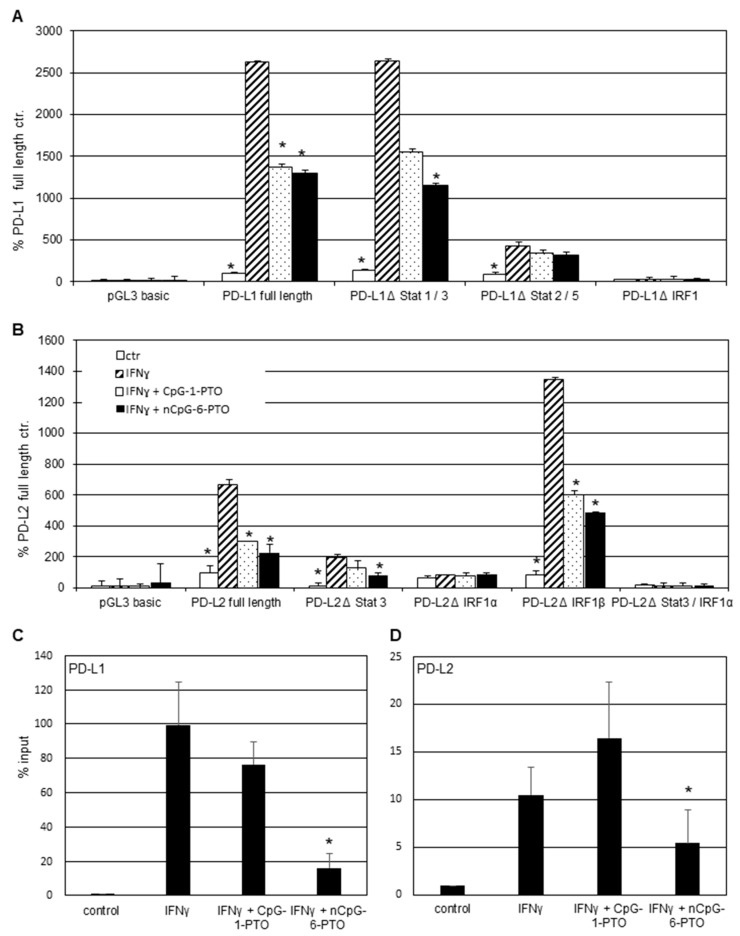
Promoter function analysis. Transient luciferase reporter assays for the (**A**) PD-L1 and (**B**) PD-L2 promoter. A375 melanoma cells were transfected with PD-L1 and PD-L2 promoter constructs including deletions of the relevant transcription binding sites. After stimulation with 20 ng/mL IFNγ for 1 h, cells were treated with 4 µM CpG-1-PTO or nCpG-6-PTO for 16 h. (**C**) ChIP assay after pretreatment with 4 µM CpG-1-PTO or nCpG-6-PTO for 1 h and consecutive IFNγ stimulation for 6 h using the IRF1 antibody for precipitation. PCR was performed with PD-L1 or (**D**) PD-L2 promoter specific primers. Each column represents the mean of 3 experiments. Statistical analysis was performed in relation to controls treated with IFNγ solely; * *p* < 0.05.

**Figure 5 cancers-14-04698-f005:**
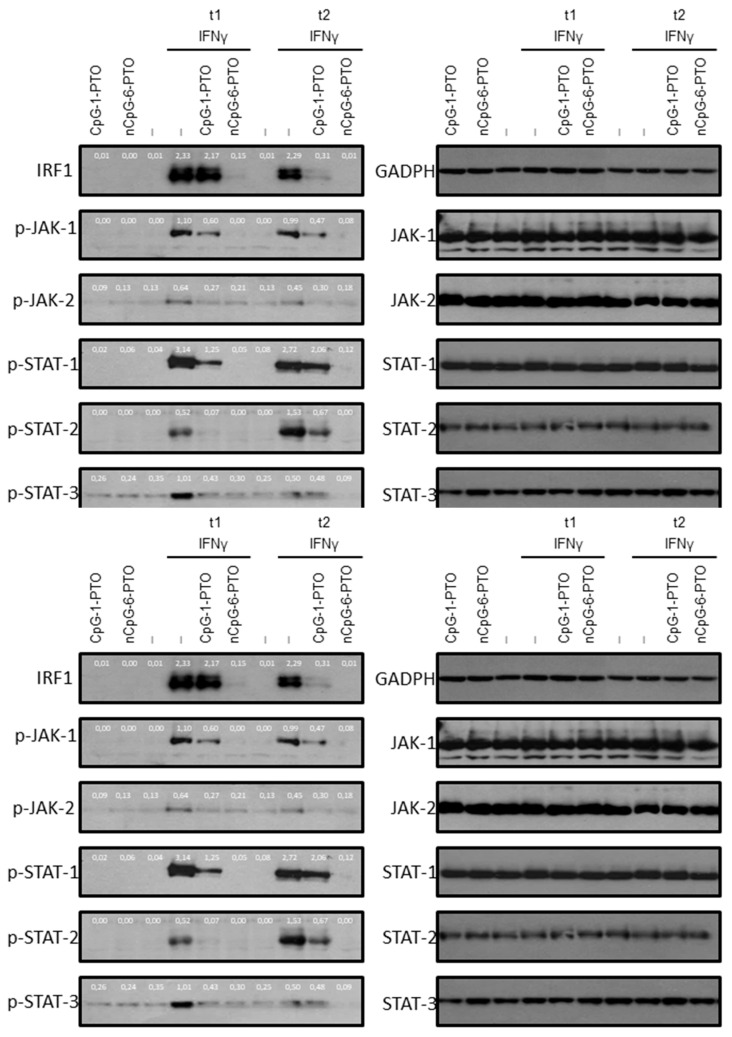
Western blot analysis of interferon receptor signaling proteins. A375 cells were treated with 4 µM CpG-1-PTO or 4 µM nCpG-6-PTO without further stimulation (basal activation, row 1–3) or with additional stimulation with IFNγ for two time periods (t1 and t2). Protein extracts were subjected to Western blot analysis and tested for expression of IRF1 (t1: 60 min, t2: 3 h), p-JAK-1 (t1: 10 min; t2: 30 min), p-JAK-2 (t1: 10 min; t2: 30 min), p-STAT-1 (t1: 10 min; t2: 30 min), p-STAT-2 (t1: 10 min; t2: 30 min) and p-STAT-3 (t1: 10 min; t2: 30 min). Equal loading was monitored using antibodies directed against the total form of the phospho-protein, or in the case of IRF1 by GAPDH (blots on the right). The blots show representative results (n = 3). p-: phospho.

**Figure 6 cancers-14-04698-f006:**
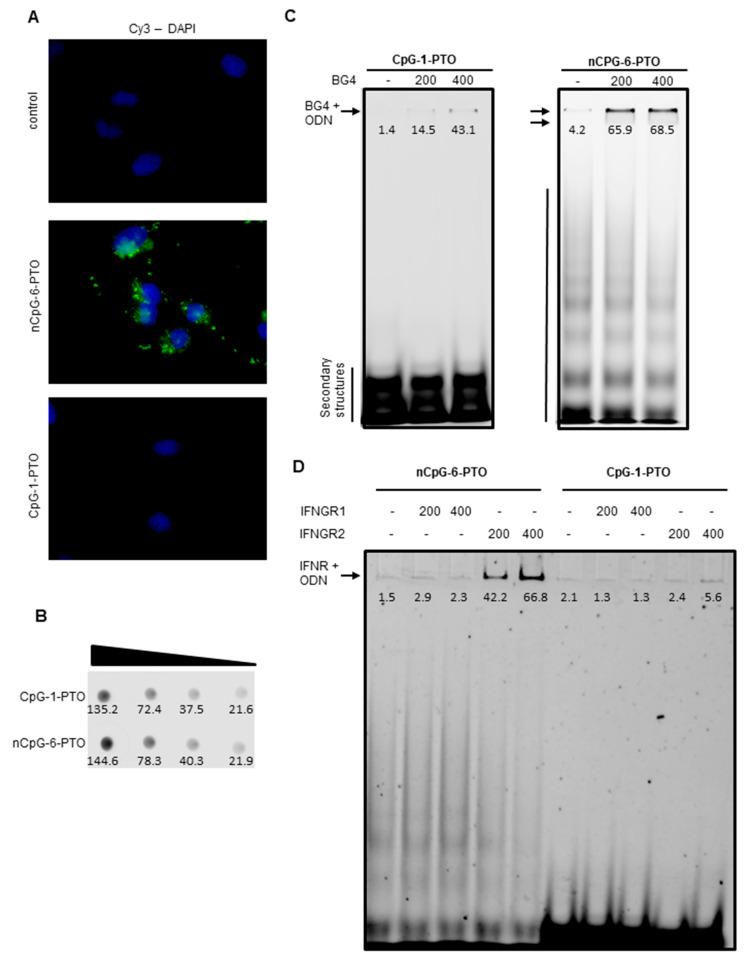
Cellular uptake of ODNs, G4 formation and binding to IFNGR. (**A**) A375 cells were treated with 4 µM 5′-Cy3-labelled CpG-1-PTO and nCpG-6-PTO for 24 h. The images show the presence of ODNs (green) in relation to the nuclei (DAPI, blue). (**B**) Dot blot analysis compared fluorescence intensities of a serial dilution (1:10) of 5‘-Cy3-labelled CpG-1-PTO and nCpG-6-PTO. (**C**) 0.2 µg Cy3-labelled CpG-1-PTO and nCpG-6-PTO were either mixed with 200 ng or 400 ng G4-binding antibody BG4 and then separated by PAGE. (**D**) 0.2 µg Cy3-labelled CpG-1-PTO and nCpG-6-PTO were either mixed with 200 ng or 400 ng IFNGR1 or IFNGR2 before separation by PAGE. The images show representative results.

**Table 1 cancers-14-04698-t001:** Oligonucleotides. Phosphorothioates (PTOs) are in capital letters, phosphodiesters (PDEs) are in small letters and CpG motifs are underlined.

CpG-1-PTO	5′-TCC ATG ACG TTC CTG ACG TT-3′
CpG-14-PTO	5′-TCC ATG ACG TTC CTG A-3′
CpG-12-PTO	5′-CAT GAC GTT CCT-3′
CpG-9-PTO	5′-GAC GTT-3′
CpG-1-PDE	5′-tcc atg acg ttc ctg acg tt-3′
CpG-1-PTO-rev	5′-AAC GTC AGG AAC GTC ATG GA-3′
nCpG-1-PTO	5′-TCC ATG AGC TTC CTG AGT CT-3′
nCpG-3-PTO	5′-TTT TTT TTT TTT TTT TTT TT-3′
nCpG-5-PTO	5′-CCC CCC CCC CCC CCC CCC CC-3′
Scrambled	5′-CTC TAG GAC TCT CTG GAC TT-3′
Oblimersen (G3139)	5′-TCT CCC AGC GTG CGC CAT-3′
nCpG-6-PTO	5′-GGG GGG GGG GGG GGG GGG GG-3′
nCpG-6B-PTO	5′-GGG GGG GGG GGG GGG G-3′
nCpG-6D-PTO	5′-GGG GGG GGG GGG-3′
nCpG-6G-PTO	5′-GGG GGG-3′

## Data Availability

Not applicable.

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
