# Peer review of "Functional Downregulation of PD-L1 and PD-L2 by CpG and non-CpG Oligonucleotides in Melanoma Cells"

_cancers, 2022, doi:10.3390/cancers14194698_

Round 1
Reviewer 1 Report
This is an important area to develop further as only a subpopulation of melanoma patients has long-term benefit from currently used checkpoint inhibitors. The authors found that a variety of different oligonucleotide (ODN) sequences significantly suppress PD-L1/2 expression and have effects on the JAK/STAT signaling pathway. They present them as interesting pharmacological compounds. However, no in vitro and/or in vivo data is included in the manuscript to demonstrate their anti-tumorigenic effect. The effect can only be studied in presence of T cells. This is essential information for the paper.
Minor:
Figure 1b. Unclear graphs, not possible to read the %.
Author Response
We like to thank the reviewer for the constructive criticisms helping us to improve the paper. Please find below our answers to the specific points raised.
We agree with critique that a functional validation of the proposed anti-tumorigenic effect of a downregulation of PD-L1 is missing. The regular mouse models are designed to check for an antibody-based interference with the PD-L1/PD-1 system and do not apply to a regulation of PD-L1 itself. Therefore we are presently working on an in vitro T-cell killing assay using T-cells from patients. We are looking forward having results within the next year.
The histograms of Figure 1B were enlarged in order to facilitate the legibility.
Reviewer 2 Report
This present article by Kleemann et al reported the importance of non-CpG oligonucleotide mediated repression in PD-L1/2. The author's claims this therapy could be a turning point for PD-L1/2 targeted therapy. I am not principally supportive of accepting this work for publication in this current format.
There is an ample amount of corrections to be made in its present form.
Major
-
Fig 1. Panel A quantification missing. Panel B FACS data was not legible.
-
Fig 2. Graph legend was missing except in Panel B, also E and F in the figure legends were mislabeled.
-
Fig5. Quantification/densitometry was missing.
-
Fig6. Scale bar was missing in panel A also its magnification was missing in figure legend.
Minor
Several spelling mistakes were detected throughout the text. IFN-Æ” was abbreviated INF in most occasions.
Author Response
We like to thank the reviewer for the constructive criticisms helping us to improve the paper. Please find below our answers to the specific points raised.
The histograms of Figure 1B were enlarged in order to facilitate the legibility.
Mislabelling of graph legend of Fig. 2 was corrected – thank you very much for pointing us to this error!
Densitometric quantification of signals from Fig. 5 and 6 was performed.
We could not find any missing γ in the naming of IFNγ. We added IFNγ under keywords.
Reviewer 3 Report
In this study, Kleemann et. al. demonstrates the TLR 9 independent inhibitory effect of solely guanines (nCpG-6-PTO) based oligonucleotide on PD-L1 expression in IFNg treated human melanoma cell lines. The conclusions reached by the authors are supported by the data presented. However, few concerns that needs to be addressed.
Major
1- The reviewer would like to know the possible effects of these ODN treatments on DCs -T cell crosstalk in the tumor microenvironment (TME). The authors should discuss this in the discussion.
2- IL27 alone and in combination with IFNg has been shown to induce the activation of STAT1 & 3 for the modulation of PD-L1 expression on human melanoma cells. The reviewer would like to see the effect ODN treatment on the IL27+IFNg induced expression of PD-L1 in A375 cell line (PMID: 31730010).
3- The authors should also show the effect of ODN treatment on p65 in Fig.5.
Minor
1- Reference missing for “Particularly, the combination of 45 BRAF- and MEK inhibitors improved overall survival in BRAF-mutated advanced melanoma and is now approved as first-line therapy”
2- The authors should include the densitometric plot with all the independent experiments done for Fig.1A and mention the number of independent experiments in the figure legend.
3- The authors finalized the 4um concentration of ODN treatment based on what experiments?
4- Typos
Line 390- A475 cells in place of A375 cells.
Line 480- Representative is written as represantive
5- Significance is missing for different experimental plots For e.g., Fig.2C significance missing for PD-L2 plots. The authors should use ‘ns’ if comparison in non-significant.
6- The authors should cite the studies defining the mutant promoter sequence for Fig.4 results.
Author Response
We like to thank the reviewer for the constructive criticisms helping us to improve the paper. Please find below our answers to the specific points raised.
1. Reference missing for “Particularly, the combination of 45 BRAF- and MEK inhibitors improved overall survival in BRAF-mutated advanced melanoma and is now approved as first-line therapy”
Answer: The work of Larkin, J., et al. "Combined vemurafenib and cobimetinib in BRAF-mutated melanoma." New England Journal of Medicine 371: 1867-1876, 2014 was added as reference.
2. The authors should include the densitometric plot with all the independent experiments done for Fig.1A and mention the number of independent experiments in the figure legend.
Answer: Experiment shown in Fig. 1A was was repeated with comparable results
3. The authors finalized the 4um concentration of ODN treatment based on what experiments?
Answer: Fig. 2 shows the concentration dependent effect of CpG-1-PTO and nCpG-6-PTO on PD-L1/2 suppression. Using 4 µM nCpG-6-PTO leads to a complete suppression of IFNg-induced PD-L1/2. As the suppressive effects of CpG-1-PTO were significantly lower a concentration range up to 8µM was tested (Fig. 2A). In ordert to compare the effect of both ODN 4µM was chosen for the following experiments.
4- Typos
Line 390- A475 cells in place of A375 cells.
Line 480- Representative is written as represantive
Answer: Both errors were corrected.
5. Significance is missing for different experimental plots For e.g., Fig.2C significance missing for PD-L2 plots. The authors should use ‘ns’ if comparison in non-significant.
Answer: Statistical significance was performed for all experiments, also for those displayed in Fig. 2C. An asterisks, as stated in the legend, indicate p-values < 0.05. The absence of an asterisk indicates no statistical significance.
6- The authors should cite the studies defining the mutant promoter sequence for Fig.4 results.
Answer: The PD-L1/2 promoter transient reporter assay was performed according tot he work from Garcia-Diaz, A., et al. ("Interferon Receptor Signaling Pathways Regulating PD-L1 and PD-L2 Expression." Cell Rep 19: 1189-1201, 2017). This work is cited under Materials and Methods and under Results (3.4. CpG-1-PTO and nCpG-6-PTO suppress PD-L1/2 mRNA and promoter activation).
Round 2
Reviewer 1 Report
A functional in vitro validation demonstrating the proposed anti-tumorigenic effect of a downregulation of PD-L1 by ODN is essential for the paper.
Author Response
The point you raise is important. As already pointed out, the numerous commercial animal models are not suitable. In these, the two interaction partners (PD-L1, PD-1) are constitutively expressed and can then be physically separated using a suitable antibody. This results in an enhanced immune response. In contrast, in our proposed mechanism, downregulation of PD-L1 (and PD-L2) occurs. Ideally, one would use a vitro model with tumor cells and T cells from the same patient. However, the hurdles for this are very high for understandable reasons. Alternatively, we are currently working on another model: A375 cells are stably transfected with pp65, a CMV antigen. In a killing assay (Eurobium release), these cells are tested with T cells from CMV-reactive patients. Oligo-dependent downregulation of PD-L1 would lead to increased killing here. The problem we currently face is that only about 1% of the subjects' T cells are specific against pp65. We are currently working on a method of enrichment. This work is very extensive and methodologically very challenging. We hope to make progress during the year and will publish the results in a separate article. We hope for your understanding. Thank you very much for your valuable comments.
Reviewer 2 Report
This current version of fig 1B still loses its legibility. Still, the fig 1B should be corrected.
IFNγ was misspelled as INFγ throughout the text, and should be corrected. I strongly encouraged the author to correct the mistake as indicated.
Author Response
- We are sorry that the readability of Fig. 1B in the version on your screen is still not satisfactory. This is probably a technical problem related to compression that we cannot fix. All I can say is that Fig. 1B is a bit small on our screen, but it is easy to see. We hope that in the final version the resolution will be higher.
- Thanks for pointing this out, I only noticed the error now. We have replaced INF with IFN throughout the text. Thank you again for your valuable advice.
Reviewer 3 Report
The authors addressed all the concerns raised by the reviewer.
Author Response
Thank you for the valuable comments to improve the article.
Round 3
Reviewer 1 Report
No further comments.
Author Response
Dear Reviewer 1,
The requested legibility of Fig. 1B was improved by increasing the resolution.
Thank you for your support!
Best,
Stefan Kippenberger